# Communication Ecology Model of Successful Aging in Indonesian Context

**DOI:** 10.3390/geriatrics8010003

**Published:** 2022-12-26

**Authors:** Irwansyah Irwansyah

**Affiliations:** Communication Department, Faculty of Social and Political Science, Universitas Indonesia, Gedung Komunikasi, Depok 16424, West Java, Indonesia; irwansyah09@ui.ac.id

**Keywords:** communication, ecology, model, successful, aging

## Abstract

The communication ecology model of successful aging (CEMSA) as a part of aging studies from a communication perspective was replicated in the Indonesian context. The CEMSA provided a specific perspective from communication scholars about the successful aging process. The study of CEMSA has grown significantly to demonstrate the importance of the interactive-communication process to propagate and enhance aging studies. However, there has been no specific aging study from a communication perspective, especially from communication scholars in the Indonesian context. This study applied all concepts, variables, measurements, and analyses from the replicated study. The result showed that seven domains of communication about aging may be relevant to predict successful aging from a negative effect and positive effect, and efficacy toward aging. The model showed that the uncertainty and combination of a negative and positive effect in seven domains of communication about aging could construct the efficacy and success of the aging process. The model with seven domains of communication about aging could be proved while the data were gathered not by self-report.

## 1. Introduction

Aging has been characterized by Plato and contemplated by Aristotle [1]. However, modern aging research emerged in the late twentieth century [2]. In 1939, the study of aging began with the documentation of the food and life expectancy of mice and rats [3]. Initially, the researchers investigated the aging rate caused by specific chronic diseases. Then, the researchers of lifespan genetics and disease models collaborated with interdisciplinary scientists [4]. One of the interdisciplinary scientists came from a communications perspective [5]. The involvement of inclusive scientific communications was beneficial to propagate and enhance aging studies. To avoid the misinformation of geroscience, the new media were used to disseminate key research findings via more unidirectional communications strategies [6]. 

The involvement of communications scientists was not only about delivering the result and the finding in aging research through different media channels and various publics [7]. One of the concerns of communications scientists was how the communications perspective became a part of aging studies. At first, communications, especially the communications function, became a part of the clinical evaluation for adult audiologic rehabilitation in institutionalized settings by Zarnoch and Alpiner in 1977 [8]. Several studies implemented the communications function as a part of aural rehabilitation in a different context such as hearing aid [9], quality-of-life changes [10], and communications strategies and attitudes for older adults [11]. However, these studies were written by non-communications scholars.

The specific aging studies from a communications scholar’s perspective started with the Handbook of Communication and Aging Research [12]. The handbook demonstrated that aging is an interactive process rather than just (or even mostly) a personal one. The collection of studies proved that the study of communications can lead people to understand what it means to grow old. The data also revealed that the negotiation of doctor–patient consultations is conducted through the communications process [13], how we perform parenthood and grandparenthood [14], and how images of aging are portrayed in the media [15]. The 22 articles discussed topics ranging from (1) the experience of aging; (2) language, culture, and social aging; (3) the communicative construction of relationships in later life; (4) organizational communications; (5) political and mass communications; (6) health communications; to (7) senior adult education [12]. 

The recent involvement of communications scholars in aging studies could be identified by several publications. As of the 27 November 2022 of this study, from the Scopus database, four studies have been published about a communications ecology model for successful aging (CEMSA). The first study focused on how young people’s perceptions of their present–future selves (present–future self-continuity) and their exposure to aging-related ambient "chatter" influenced how they felt about older people as well as how effective they were at dealing with aging-related anxiety and efficacy [16]. The second research paper examined the relationship between young adult children’s negative sentiments and perceptions of their own efficacy towards aging and the frequency of their parents’ unfavorable discussions about aging (i.e., complaints about age and use of age-based excuses) [17]. The third research paper looked at how the themes of age-related communications and memorable messages of the elderly were indirectly connected to successful aging through aging effectiveness, and how these themes were related to older adults’ dispositional hope [18]. The fourth study expanded the communications ecology model of effective aging by investigating the impact of professional purpose and retirement planning on American employment [19].

The expansion of communications and effective aging research including CEMSA as a new perspective from communications scholars has significantly increased. From the Scopus database, 21 articles about communications and successful aging were published and indexed from 2015–2022. However, in the Indonesian context, the publications related to aging in 2022 were about health [20,21,22,23], quality of life [24,25], construction of policy [26,27], public bathroom design [28], macro-demographics [29], gross saving [30], somatosensory stimulation [31], intergenerational support [32], and pension [33,34]. As of 27 November 2022, there was no specific aging study from a communications perspective, especially from communications scholars. Therefore, the purpose of this study was to address the lack of communications research on aging in Indonesia. This study replicated an updated CEMSA due to its growth in communications perspective and lack of communications study focusing aging in the Indonesian case. A new and improved CEMSA [35] suggests that (un)certainty about and other aspects of the aging process may be anticipated by ambient chatter, which in turn encourages communicative engagement in other domains. The expanded and improved CEMSA model has not been exhaustively examined in a single report [36]. The recent studies of CEMSA recommended future research to explore the continuity influences for the aging experiences of individuals [16,19]. 

The Communication Ecology Model of Successful Aging (CEMSA) was established to expand the role that communication plays in the aging process [37]. CEMSA was developed based on the idea that a person’s assessment of uncertainty about age and aging is shown by the way a person communicates. Communication as a part of interaction [38] predicts successful aging [39]. When people communicate about their age and aging in higher uncertainty it leads to more negative effect [36]. In reverse, people’s communications about their age and aging in lower uncertainty led to a more positive effect [36]. A related study indicated that more uncertainty and negative mood were related to diminished efficacy, whereas greater positivity was related to increased efficacy [36].

The initial model consisted of the concepts of (1) uncertainty, (2) positive effect and negative effect about aging, (3) communication about aging, (4) efficacy, and (5) successful aging [37]. Uncertainty refers to a situation that is ambiguous and unpredictable [40]. Uncertainty occurs when there is inconsistent or unavailable information that makes people insecure in their own knowledge [41]. Uncertainty comes when someone assesses an unfamiliar event [42]. Uncertainty is multilayered [41]. The first layer is about the self of belief, values, abilities, and behaviors. The second layer is about the other of belief, values, abilities, and behaviors. The third layer is about quality and durability of relationships. The other layer is about the features of context such as rules, social norms, and procedures [41]. Uncertainty about aging relates to the difference between experiences and desire [36] that could be a portion of negative experience perceived probability [35] associated with getting older [43].

Positive effect and negative effect about aging refers to effective reason [37] or psychological motive well-being [44]. Positive effect and negative effect indicate distinct emotional or expressive elements of subjective happiness. The positive effect refers to the degree to which something of an individual feels awake, energetic, and excited. Great positive effect is characterized by total concentration, pleasurable participation, and extreme vitality, whereas low positive effect is characterized by fatigue and sadness [45]. The negative effect is the degree to which a person feels remorse, indignation, contempt, and dread [46]. Low negative impact is a sense of peace and tranquility [45].

Communications regarding aging refers to a range of communication actions associated with age and aging [47]. Communication plays an important part in the social and communicative development of the age and aging environment [37]. Communication about aging refers to interactions between persons in which they discuss various facets of the aging process [19]. Communication about aging consist of seven domains: (1) Self-categorization as old, or attributing behavior to old age; (2) Expressing optimism about the aging process; (3) Collusion in categorizing and teasing others; (4) Resisting mediated images of age and peddlers of anti-aging products; (5) Planning for future care needs; (6) Use of emerging communication technologies; and (7) Managing being the recipient of ageism [37,47]. 

Efficacy refers to a person’s belief that someone can accomplish a desired result [39]. Aging efficacy is a measure of how well a person thought they could handle problems that might come with getting older [37]. Aging efficacy pertains to a person’s adaptability to aging. The notion of aging effectiveness is associated with conquering the challenges of aging. Individuals’ opinions of their capacity to regulate the aging process are referred to as their efficacy about aging [36]. Aging efficacy refers to the confidence that individuals have in their ability to cope with the impacts of aging [19]. 

Successful aging is described as individuals’ subjective assessments of how well they are aging [47]. Successful aging means individuals’ judgments of how well they were aging [37]. Successful aging refers to an individual’s perception of how well they are coping with the aging process [19]. Validity of a two-factor model of successful aging was shown by the fact that successful aging may be described by two distinct factors involving objective and subjective success [48]. Older adults describe successful aging by endorsing social engagement and a positive outlook toward life [49]. Further replication was made based on the previous hypothesis presented in Table 1. 

CEMSA presents a paradigm that relates uncertainty about aging to age-related communications, emotional reactions to aging, aging-related effectiveness, and finally successful aging. According to the CEMSA, uncertainty about aging has the ability to elicit emotional reactions to aging and impact the inclination of persons to participate in a number of communications activities related to age and aging. Together with uncertainty and emotional reactions to aging, it is believed that these age-related communications activities generate ecologies that assist or obstruct effective aging by influencing the degree of efficacy people feel towards the aging process. In consequence, it is anticipated that perceived effectiveness about aging will function as a direct antecedent of successful aging itself (See Figure 1).

## 2. Methods and Measure

This research replicated previous studies [16,17,36,37,43] of CEMSA in the Indonesian context. Indonesian law no. 13/1998 on the welfare of the elderly states that someone could be elderly for someone who has reached the age of 60 years and over [50]. Most of the CEMSA studies were tested on participants or respondents between 40 years old and over 65 years old [17,19,36,37,43,47]. The 618 respondents of a random sample came from a dataset of three cities that have more than 14 percent of elderly people [51]. After obtaining institutional review board approval (380/VIII/2022/KEPK), the respondents were asked by resident enumerator assistance to fill the questionnaire. The resident enumerators were chosen due to their familiarity with local customs and dialects [52]. The instrument was translated from English to Bahasa Indonesia with several revised words and sentences based on local understandings and contexts. Data collection was carried out for three months from June–August 2022. Participants in this research received either a modest monetary reward (equivalent to USD 5) or a box of groceries such as rice, sugar, and cooking oil.

This study measured all variables of the original CEMSA that consist of (1) demographic profile, (2) uncertainty, (3) positive effect and negative effect, (4) communication about aging, (5) efficacy, and (6) successful aging (See Table A7). The demographic profile was explained from sex, age, ethnicity, education, residence status, marriage status, employment, and support source. Similar to previous studies [37,47], the demographic profile would be associated with other variables of CEMSA.

In terms of demographic profile, the majority of 618 respondents (62%) were female, while the remainder (38%) were male. The average age was 62.3 years old (SD = 3.31 years). Most of the respondents were Javanese (99.5%) since the three cities were on Java Island. The other respondents had Bataknese (0.3%), Madurenese (0.1%), and Chinese (0.1%). The last education of respondents was identified as senior high school (79%), higher education (13%), junior high school (12%), and primary school (6%). Residence status of respondents was living with own kids (83%), living with big extended family (12%), and living alone (5%). The respondents were currently married (68.2%), the remaining being widowed (31.8%). Majority of respondents were retired (88%), not working (6%), still full-time working (4%), and part-time working (2%). To fulfill their daily basic need, the respondents received support from pension fund (88%), their children (12%), paid salary (6%), and extended families (4%). 

Uncertainty was measured by three statements such as (1) "I am less certain than I would like about my future"; (2) "I know less than I would like about what my life will be like as I age"; and (3) "I don’t know as much as I would like about how growing older will feel" [37,47]. Respondents were asked to rate their agreement or disagreement on a seven-point scale: (1) strongly disagree, (2) disagree, (3) slightly disagree, (4) neither agree nor disagree, (5) somewhat agree, (6) agree, and (7) strongly agree [53]. Table A1, Table A2 and Table A3 displays the means, standard deviations, alpha reliability, and correlations for each composite measure.

Five positive and four negative emotions were used to evaluate positive and negative effect considering respondents’ feeling about aging [19]. Five positive emotions are (1) excited, (2) interested, (3) strong, (4) inspired, and (5) enthusiastic, and four negative emotional states are (1) nervous, (2) scared, (3) distressed, and (4) upset [47]. Nine items of emotions were similar to previous studies [37,44] that validated from Positive Effect and Negative Effect Schedule (PANAS) [54]. On a 5-point scale, respondents were asked how they felt about aging: (1) very barely/not at all, (2) slightly, (3) moderately, (4) quite a bit, and (5) very considerably [19,37].

Communications about aging were assessed based on 21 criteria that aimed to represent the manner in which communication generates or alters ecologies of aging [37]. The 21 items were categorized into seven domains [47]. The first domain was about self-categorization as old, or attributing behavior to old age. This domain had three items: (1) “When I forget something, or have trouble with a task, I often say it is because of my age”; (2) “I often hear myself explaining away some event by referring to my age”; and (3) “When I talk about what is happening in my life, I frequently mention my age”. The second domain was expressing optimism about the aging process. The domain had two items: (1) “Frequently express the fact that I am optimistic about aging” and (2) “When I talk about my age or that I am aging, I often sound sad about it”. The third domain was about collusion in categorizing and teasing others. The domain had three items: (1) “When telling stories about myself and my life, I frequently refer to the joys and rewards of being older”; (2) “I often tease others about their age”; and (3) “When I send birthday cards or messages, I often poke fun at the person’s age”. The fourth domain was about resisting mediated images of age and peddlers of anti-aging products. The domain had three items: (1) “I look for and purchase ‘antiaging’ (e.g., skin creams; hair restoration) products I read about in magazines and/or see on television”; (2) “I resent the ads and products that claim I should work at looking younger”; and (3) “I’m skeptical when I see an ad for a product that will help me ‘recapture my youth’”. The fifth domain was about planning for future care needs. The domain had three items: (1) “I have talked with my family and friends about my wishes regarding care as I age”; (2) “I’ve tried to make sure my family are informed about my preferences if they ever need to make health decisions for me”; (3) “I have talked with my older family members about their wishes regarding care as they age”. The sixth domain was about use of emerging communication technologies. The domain had three items: (1) “I enjoy keeping up with new communication technologies such as social media and smart-phone apps”; (2) “I find communicating via e-mail, Skype, and other electronic channels frustrating”; and (3) “I encourage friends and family members to make use of new communication technologies”. The item of “I find communicating via e-mail, Skype, and other electronic channels frustrating” was adapted into an Indonesian context. Skype as electronic communication channels was replaced by WhatsApp since it was used by 121 million Indonesian users [55]. The adapted item became “I find communicating via WhatsApp, email, and other electronic channels frustrating”. The seventh domain was about managing being the recipient of ageism. The domain had four items: (1) “I often make jokes about someone’s age if they are having problems with some tasks or other”; (2) “When others make jokes about my age, I usually play along”; (3) “I normally have a comeback for people who make inappropriate comments or tease me about my age”; and (4) “When people talk to me as if I am just an older person, it bothers me”. The 21 statements were measured using 7 Likert-type forms ranging from strongly disagree, disagree, slightly disagree, neither agree nor disagree, somewhat agree, and agree to strongly agree [39].

Efficacy was assessed by five items about individual’s belief, feeling, perception or ability to cope with aging process [37,39,47]. The five items were about (1) “I feel that I can deal with any challenges that growing older will bring”; (2) “I feel able to cope with things that might happen to me as I age”; (3) “I am confident that if I need to, I will be able to adapt to age-related changes”; (4) “I feel fully in control of dealing with my own aging”; and (5) “I am not in command as much as I should be about growing older” [36]. The scale was calibrated to seven points: Strongly disagree, disagree, slightly disagree, neither agree nor disagree, somewhat agree, agree, and strongly agree on a Likert-type scale [37].

Six items from participants’ assessments of how well they were aging were used to determine successful aging [47]. The six items were about (1) “How successfully have you aged up to now?”; (2) “How well are you aging?”; (3) “How do you rate your life these days?”; (4) “I am happy with the age I am right now”; (5) “At my age I feel that life has much to offer”; and (6) “I’m as happy at this stage of my life as I have been at other points in time” [37]. The first of three items was responded on a 7-point semantic differential scale (not at all well, low well, somewhat well, neutral, moderately well, well, and very well), while the next three questions were answered on a 7-point Likert scale (strongly disagree, disagree, somewhat disagree, neither agree or disagree, somewhat agree, agree, and strongly agree) [39].

All variables for the communications ecology model of successful aging were reliable (α = 0.83–0.99; *M* = 1.00–2.40; SD = 0.66–2.18). Especially the variables of communication about aging, the 21 items that categorized into seven domains were also reliable for measured into a single composite variable (α = 0.86; *M* = 1.79; SD = 1.78). This study found the seventh domain of communications about aging, managing being the recipient of ageism that contained four elements, dependable (α = 0.86; *M* = 2.40; SD = 1.85). Previous research employed just a single unreliable item and a face-valid measure of ageism management competence [37]. This study proved that managing being the recipient of ageism from previous study was relevant as a part of the communications about aging variable. 

## 3. Results and Discussion

The hypotheses as a part of CEMSA were tested through a structural equation model in AMOS 26. Similar to a previous study, this study employed computing by employing a maximum likelihood estimate for each of the aging-related communications [37]. The result of each computation was equipped by χ2 statistics, the standardized root mean square residual (SRMR), root mean squared error of approximation (RMSEA), and comparative fit index (CFI). The good fit model required lower or <2 of χ2 statistic [56], a cut-off of ≤0.08 score for SRMR and RMSEA < 0.6 [57,58], and CFI > 0.95 [59]. See Table A4 about model fit indices.

The first hypothesis was the correlation between uncertainty relating to aging and the seven domains of communication about aging. The result based on standardized regression coefficients supported the first hypothesis (H1). The greater the degree of uncertainty around aging, the more individuals prepared for future care requirements (β = 0.68, *p* < 0.01); the greater the proportion of individuals who self-classify as elderly or ascribe conduct to old age, the greater was the correlation between self-classification and behavior (β = 0.67, *p* < 0.01); the more participants colluded in categorizing and teasing others (β = 0.67, *p* < 0.01), the more people experienced age discrimination (β = 0.58, *p* < 0.01); the more optimistic people were about the aging process, the greater was the utilization of developing communications technologies by participants (β = 0.48, *p* < 0.01); and the more individuals fought age media and anti-aging product vendors, the greater was their resistance. This study showed that all limited support from a previous study [47] could be removed for Indonesian context.

The second hypothesis (H2) stated that uncertainty about aging is associated with more negative and less positive feelings about aging. The finding of this study was that the uncertainty amplified adverse effects (mean β = 0.30, *p* < 0.01) and lowered positive influences (mean β = −0.17, *p* < 0.01). This finding was similar to the previous study [47] that supported the H2.

The third hypothesis (H3) said that a negative effect and ambiguity about aging are negatively associated with aging efficacy, whereas a good effect toward aging is favorably associated with aging efficacy. This study revealed that uncertainty was negatively related to effectiveness (mean β = −0.20, *p* < 0.01). Moreover, the negative effect regarding growing older predicted reduced efficacy about aging (mean β = −0.29, *p* < 0.05), and the favorable effect for aging indicated a rise in aging effectiveness (mean β = 0.40, *p* < 0.01). The findings support the H3 that is similar from the previous studies [19,47].

The fourth hypothesis (H4) stated that the more people (1) self-categorize as old, or attribute behavior to old age; (2) express optimism about the aging process; (3) collude in categorizing and teasing others; (4) resist mediated images of age and peddlers of anti-aging products; (5) plan for future care needs; (6) use emerging communication technologies; and (7) manage being the recipient of ageism, the more they report negative effects with respect to aging, and the less they report positive effects and efficaciousness about aging. Self-categorization as old or attributing behavior to old age (β = 0.51, *p* < 0.01), collusion in categorization and teasing others (β = 0.51, *p* < 0.01), use of emerging communications technologies (β = 0.45, *p* < 0.01), and planning for future care needs (β = 0.28, *p* < 0.01) each predicted higher levels of positive effects about aging. Meanwhile, expressing optimism about the aging process predicted lower levels of positive effects about aging (β = −0.29, *p* < 0.01) and a higher level of negative effects about aging (β = 0.25, *p* < 0.01). This study also found that self-categorization as old or attributing behavior to old age (β = −0.25, *p* < 0.01), collusion in categorizing and teasing others (β = −0.25, *p* < 0.01), planning for future care needs (β = −0.10, *p* < 0.01), and use of emerging communications technologies (β = −0.10, *p* < 0.01) predicted a lower level of negative effects towards aging. The finding of this study was similar to the previous study [47] that had mixed support for H4.

In terms of efficacy, collusion in categorizing and teasing others (β = 0.79, *p* < 0.01), self-categorization as old or attributing behavior to old age (β = 0.68, *p* < 0.01), planning for future care needs (β = 0.56, *p* < 0.01), managing being the recipient of ageism (β = 0.25, *p* < 0.01), resisting mediated images of age and peddlers of anti-aging products (β = 0.22, *p* < 0.01), use of emerging communications technologies (β = 0.14, *p* < 0.05), and expressing optimism about the aging process (β = 0.12, *p* < 0.05) predicted a higher level of efficacy about aging. This study supported all domains of communication about aging related to efficacy and supported H4. This finding was different from a previous study [47] that had mixed support for H4.

The fifth hypothesis (H5) stated that efficacy is positively associated with successful aging. This study supported H5 and found that there was a strong relation between efficacy about aging and successful aging in every communication-about-aging domain (mean β = 0.97, *p* < 0.01). The finding was similar to previous studies [19,37]. This study found a higher prediction than the previous studies that had mean β = 0.85, *p* < 0.001 [37] and mean β = 0.71, *p* < 0.001 [19].

In terms of the sixth hypothesis (H6) that stated communication has an indirect effect on successful aging via effect and efficacy, this study partially supported H6. Collusion in categorizing and teasing others (β = 0.69, *p* < 0.001), planning for future care needs (β = 0.54, *p* < 0.001), managing being the recipient of ageism (β = 0.40, *p* < 0.01), resisting mediated images of age and peddlers of anti-aging products (β = 0.21, *p* < 0.01), expressing optimism about the aging process (β = 0.13, *p* < 0.01), and using emerging communications technologies (β = 0.13, *p* < 0.01) had a significant indirect effect on successful aging via efficacy. Using emerging communications technologies (β = 0.29, *p* < 0.01), collusion in categorizing and teasing others (β = 0.10, *p* < 0.001), planning for future care needs (β = 0.10, *p* < 0.001), and managing being the recipient of ageism (β = 0.10, *p* < 0.01) had a significant indirect effect on successful aging via a positive effect and efficacy about aging (See Table A5 and Table A6). This finding was quite similar to a previous study [37] that supported H6 partially. 

As a replication study, the results showed the CEMSA model was fit and similar to a previous study [37], including support of all the hypotheses (See Figure 2). This study supported the model that showed communications was central for developing successful aging. As a determinant key, communications played an important role for any person to have a better growing-old and shape-aging process [19]. The fit model showed that the uncertainty about aging decreased a positive effect, increased a negative effect for seven domains of communications ecology about aging. Five domains of communications ecology decreased negative effects about aging and six of seven domains of communications ecology increased positive effects about aging. All domains of communications ecology as predicted and positive effects toward aging also increased efficacy about aging. Similar to a previous study [37], negative effects toward aging decreased efficacy about aging. Finally, the model also showed that the efficacy about aging increased the successful aging.

In Indonesian contexts, elderly people may face uncertainty [60]. The uncertainty feeling of older people could be detected from the changing daily behavior to see their future [61]. Elderly people reduced their uncertainty by establishing friendship relationships and communicating among them [62]. Relationships and communications helped people reduce uncertainty [63]. This study confirmed that the higher uncertainty about aging, the more people need all domains of communications about aging. Uncertainty reduction was a vital concern for the conduct of almost any communicative transaction [64]. The conceptualization of communications could be a resource to cope with uncertainty [43].

Similar to a previous study [37], this study found that uncertainty toward aging could increase the negative effect and decrease the positive effect about aging. Almost every aspect of our lives is permeated with uncertainty, including aging. In the majority of circumstances, uncertainty is connected with a negative effect, but in other instances, it is associated with a positive effect [65]. A negative effect becomes more negative and a good effect becomes more positive as a result of uncertainty [66]. It is also in line with a previous study that found that uncertainty heightens a negative effect and dampens a positive effect [67]. However, uncertainty could change from a negative effect to a positive effect such as on the life adaptation [68]. Uncertainty regarding a pleasant occurrence attracts people’s attention, encouraging them to analyze the event’s possibility, which raises their positive effect [69].

Efficacy toward aging was found to be correlated with both negative and positive ambiguity about aging. This finding was similar to a previous study that uncertainty that increased a negative effect caused the reducing of efficacy about aging, while the uncertainty that decreased a positive effect caused the increasing of efficacy toward aging [37]. Negative and positive effect were a significant moderating aspect in the relationship between uncertainty and efficacy. However, among the seven areas of communications about aging, only the collaboration in identifying and teasing others regulating negative and positive effect could not be a moderating factor between uncertainty and effectiveness. The indicators of collusion in categorizing and teasing others (such as “when telling stories about myself and my life, I frequently refer to the joys and rewards of being older”; “I often tease others about their age”; and “when I send birthday cards or messages, I often poke fun at the person’s age”) may provide a different context among uncertainty, negative and positive effect, and efficacy about aging.

As a replication study toward the CEMSA, this study proved that seven domains of communication about aging may be relevant to predict successful aging from negative effect, positive effect, and efficacy toward aging. A previous study showed that the significant indirect effect via the combination of effect and efficacy only occurred in three domains of communication about aging [37]. The difference in the finding on the domains of communication aging could be related to the adaptation of a research instrument for an Indonesian context. The indicators of the seven domains of communication about aging were adapted through the language [70] and culture style [71]. 

This study supported the conceptualizations of successful aging to emphasize the avoidance of disability and high levels of physical functioning, including communications ability as requirements for well-being [72]. The previous study found that late adulthood cognitive function could be activated via communications to examine the mental well-being channel [73]. The physical and communication behavioural modifications can lead to linked improvements in both mental and cognitive wellbeing for older adults [74]. The communications ecology model may be relevant to an age-friendly community [75]. A previous study also showed that older adults with greater use of adaptation via communications may cope and have defense mechanisms across adulthood [76]. 

A previous study showed that only three of the seven communicative behaviors studied were found to predict indirectly successful aging via their relationships with positive effect, negative effect, and efficacy [37]. The study applied an online survey to complete the measurement. In contrast, the current study applied resident enumerators to assist the respondents in filling out the questionnaire [77]. Applying the resident enumerators built trust between enumerators and respondents. Similar to another previous study, the current study showed the important of a unique study assistant using a standardized questionnaire [78]. As a result, the communications ecology model of successful aging for Indonesian context may support previous studies [17,18,19,35,36,37,39,43,79] that have consensus on the importance of communications as a central and important factor to face the aging process.

## 4. Conclusions

A replication study of the communications ecology model of successful aging in an Indonesian context became a part to demonstrate that communications was playing an important role to predict the efficacy toward aging and successful aging. The model showed that the uncertainty, in combination with negative and positive effects in seven domains of communication about aging, could construct the efficacy and successful aging process. The seven domains of communication about aging had a significant indirect effect on successful aging through positive effect and efficacy about aging. 

The fit model with seven domains of communication about aging could be proved while the data was gathered not by self-report. The assistance by resident enumerators on filling the instrument for older people above 60 years old may provide better data to confirm the real experience and situation. At the same time, resident enumerators build trust between enumerators and respondents. A future study could document and analyze any comments and responses from the assisted older people qualitatively with special communication topics about the aging process.

## Figures and Tables

**Figure 1 geriatrics-08-00003-f001:**
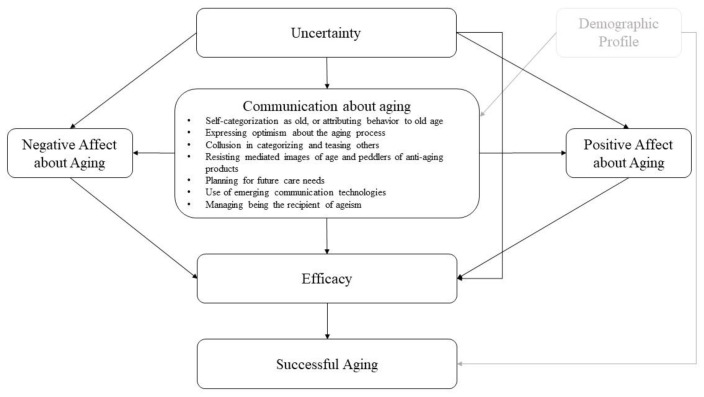
The original of the communication ecology model of successful aging (CEMSA) [37].

**Figure 2 geriatrics-08-00003-f002:**
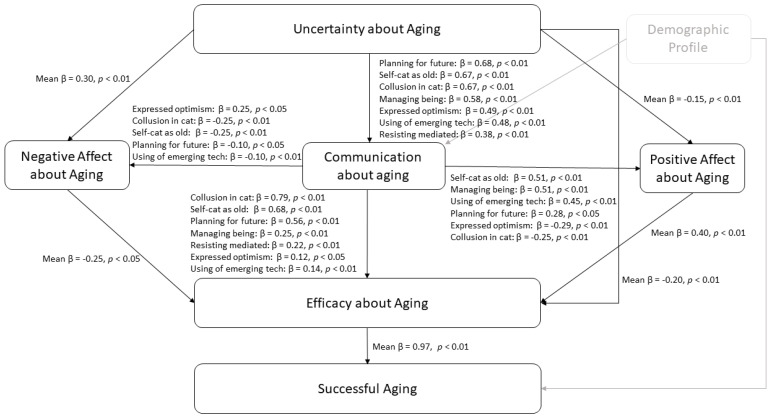
Results of Hypothesis 1 (H1)—Hypothesis 5 (H5).

**Table 1 geriatrics-08-00003-t001:** The existing hypothesis based on previous study of original CEMSA.

Hypothesis	Statements
H1	The more uncertainty about aging participants report, the more they (1) self-categorize as old or attribute behavior to old age; (2) express optimism about the aging process; (3) collude in categorizing and teasing others; (4) resist mediated images of age and peddlers of anti-aging products; (5) plan for future care needs; (6) use emerging communication technologies; (7) manage being the recipient of ageism [37].
H2	Uncertainty about aging predicts more negative and less positive emotions about aging [19,37].
H3	Uncertainty about aging and a negative effect regarding growing older are inversely related to aging efficacy, whereas a positive effect toward aging is positively related to aging efficacy [19,37].
H4	The more people (1) self-categorize as old or attribute behavior to old age; (2) express optimism about the aging process; (3) collude in categorizing and teasing others; (4) resist mediated images of age and peddlers of anti-aging products; (5) plan for future care needs; (6) use emerging communication technologies; (7) manage being the recipient of ageism, the more people express a negative effect toward aging, the less they reflect a positive effect and efficacy towards aging [37].
H5	Efficacy is positively associated with successful aging [19,37].
H6	Communication has an indirect effect on successful aging via effect and efficacy [37].

## Data Availability

All study data are included in the present manuscript.

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
