# Peer review of "Communication Ecology Model of Successful Aging in Indonesian Context"

_geriatrics, 2022, doi:10.3390/geriatrics8010003_

Round 1
Reviewer 1 Report
Overall, the study engages an important topic against the backdrop of a global ageing crisis. The study is well-angled and well-written, but there are several improvements needed before it can be accepted.
- The study lacks a concise summary of all the dimensions underlie CEMSA. Would you expect weighting heterogeneity within/across dimensions? and why not?
- What are the limitations of using self-rated wellbeing/affect measures? This is a major concern in the current study, since self-ratings are highly subjective and could be influenced by environment/ community factors.
- I randomly entered the given keywords in the WoS database. It is appropriate to cite e.g.
Article 1: https://www.webofscience.com/wos/alldb/full-record/WOS:000800848600001
Article 2: https://www.webofscience.com/wos/alldb/full-record/WOS:000339878300030
Article 3: https://www.webofscience.com/wos/woscc/full-record/WOS:000856037300001
Article 4: https://www.webofscience.com/wos/alldb/full-record/WOS:000816577600001
These articles point to the link between physio-/psycho-social factors and healthy ageing. I believe that this need to be added to the current paper, where the authors miss a discussion on why CEMSA matter and shed light on the underlying mechanisms through which CEMSA might lead to more successful ageing among older adults.
Author Response
Thank you very much for your review of the manuscript.
I would like to answer several questions regarding your questions:
1. The study lacks a concise summary of all the dimensions underlie CEMSA. Would you expect weighting heterogeneity within/across dimensions? and why not?
My answer: (1) I added a paragraph about a concise summary of all dimensions of CEMSA on page 4 of 21, and I also added table A7 about Items of Variables from Communication Ecology Models of Successful Aging before the reference list. In terms of weighting heterogeneity within/across dimensions, I am not quite sure that I could answer your question. My study was a replication study with a different context from the previous study. The previous study did not explain weighting heterogeneity within/across dimensions.
2. What are the limitations of using self-rated wellbeing/affect measures? This is a major concern in the current study, since self-ratings are highly subjective and could be influenced by environment/ community factors.
My answer: I added the academic argumentation about the importance of the resident enumerator on page 10 of 21. I hope the argumentation could answer your question.
3. I randomly entered the given keywords in the WoS database. It is appropriate to cite e.g.
Article 1: https://www.webofscience.com/wos/alldb/full-record/WOS:000800848600001
Article 2: https://www.webofscience.com/wos/alldb/full-record/WOS:000339878300030
Article 3: https://www.webofscience.com/wos/woscc/full-record/WOS:000856037300001
Article 4: https://www.webofscience.com/wos/alldb/full-record/WOS:000816577600001
These articles point to the link between physio-/psycho-social factors and healthy ageing. I believe that this need to be added to the current paper, where the authors miss a discussion on why CEMSA matter and shed light on the underlying mechanisms through which CEMSA might lead to more successful ageing among older adults.
My answer: Thank you very much for suggesting additional references to give the discussion of my study better. I added them into paragraphs on page 9 of 21 and page 10 of 21. Please let me know if any reviews on them. Thank you.
The attached file is my revised manuscript with track changes and additional academic argumentation on it.

Reviewer 2 Report
The article is very good but I think one would need to draw in the relevant references into the conclusion and reiterate the points raised in the paper.
Author Response
Thank you very much for reviewing my manuscript.
I would like to address comments and suggestions on it.
The article is very good but I think one would need to draw in the relevant references into the conclusion and reiterate the points raised in the paper.
My response: I added two new paragraphs in the conclusion that reiterate the point raised in the paper on page 10 of 21 in the conclusion section.
The attached file is my revised manuscript. Please let me know if there is any additional review for the manuscript. Thank you.

Round 2
Reviewer 1 Report
Acceptable